

# Efficient on-shell matching

**Mikael Chala**[⋆]**, Javier L. Miras**[†]**, José Santiago**[‡] **and Fuensanta Vilches**[∘]

Departamento de Física Teórica y del Cosmos, Universidad de Granada,
Campus de Fuentenueva, E-18071 Granada, Spain

⋆ mikael.chala@ugr.es , † jlmiras@ugr.es , ‡ jsantiago@ugr.es , ∘ fuenvilches@ugr.es

## Abstract

We propose an efficient method to perform on-shell matching calculations in effective field theories. The standard off-shell approach to matching requires the use of a Green's basis that includes redundant and evanescent operators. The reduction of such a basis to a physical one is often highly non-trivial, difficult to automate and error prone. Our proposal is based on a numerical solution of the corresponding on-shell matching equations, which automatically implements in a trivial way the delicate cancellation between the non-local terms in the full theory and those in the effective one. The use of rational on-shell kinematics ensures an exact analytic solution despite the numerical procedure. In this way we only need a physical basis to perform the matching. Our procedure can be used to reduce any Green's basis to an arbitrary physical one, or to translate between physical bases; to renormalize arbitrary effective Lagrangians, directly in terms of a physical basis; and to perform finite matching, including evanescent contributions, as we discuss with explicit examples.

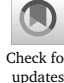

# 1   Introduction

Effective field theories (EFTs) have become the standard language in the search of physics beyond the Standard Model (SM). In the presence of a mass gap between the scale of new physics and the one that experiments are currently probing, EFTs provide a model-independent parametrization of experimental data in the form of global fits [1–10]. Indeed, all experimental observables can be described in terms of a finite number of Wilson coefficients (WCs), the couplings of local operators in the effective Lagrangian. A minimal set of operators needed to parameterize experimental observables at a given order in the EFT power counting is called a physical basis. Some phenomenologically relevant examples include the Warsaw basis [11] for the Standard Model EFT or SMEFT (see [12,13] for recent reviews) at mass dimension 6, the counterpart with right-handed neutrinos or NSMEFT [14–18], the low-energy versions of these, in which the top quark, the $W$, $Z$ and Higgs bosons are integrated out, named respectively LEFT [19] and NLEFT [20–23], as well as the EFT for the SM extended with axion-like particles [24–27]. Bases for operators of these theories at even higher dimension are also known [28–34].

The model-independent global fits can then be used to extract information on specific new physics models by matching them onto the EFT. This matching consists in computing the WCs of the EFT in terms of the parameters (couplings and masses) of the new physics model, that we will refer to as *full* model hereafter, and it can be performed either functionally or diagrammatically. In the functional approach, heavy physics is integrated out directly in the path integral, leaving a non-local effective action that can be power expanded in local operators. The resulting operators are in general not in a physical basis, and they have to be reduced to the physical basis by means of field redefinitions [35,36] (equivalent to the application of the equations of motion at the linear level). This process, while straightforward in principle, can become quite tedious and error prone at higher orders. It can be automated, as done in `Matchete` [37], although it currently only allows for Warsaw-like bases and it is non-trivial to develop a robust algorithm that allows for a free choice of physical basis by the user.

The alternative, diagrammatic approach to matching has been recently fully automated in `MatchMakerEFT` [38]. Two different paths can be taken here. The first is performing an off-shell matching, in which one-light-particle-irreducible (1lPI) Green's functions are computed in the full model and in the EFT for arbitrary off-shell kinematics. Their difference, when expanded in the heavy scales, is local and can therefore be matched by local operators in the EFT. This approach, which is the one currently implemented in `MatchMakerEFT`, has many advantages, including the fact that only a relatively small (1lPI) number of diagrams has to be computed; that the hard region contribution of the full model is directly local and corresponds to the matching contribution; and that the off-shell kinematics provide a significant amount of redundancy that can act as a very efficient cross-check of the calculation. The main disadvantage is that the basis of EFT interactions, denoted as *Green's basis*, must involve redundant operators to parameterize the off-shell Green's functions. These operators are equivalent to those in a physical basis for any observable and can therefore be reduced via field redefini-

tions as in the functional approach, this being equally inconvenient.[1] Furthermore, if only gauge invariant operators are to be included in the Green's basis, then the background field method [39] must be used, entailing some extra complications.

The second and alternative possibility within the diagrammatic approach is to perform the matching on-shell [40]. On-shell matching requires the calculation of all connected, amputated Green's functions, including light bridges (tree level propagators of light particles connecting two different parts of the corresponding diagram) in the full model and in the EFT. External particles must be put on-shell and be dressed with the corresponding factors of the wave-function renormalization constant. The great advantage of this approach is that there is no need to ever use a Green's basis, as all the physical observables can be parameterized directly in terms of the physical operators. Furthermore, any physical basis can be used, not being tied to the one enforced by the corresponding reduction algorithm. There are some disadvantages that have prevented the development of an automated on-shell matching algorithm so far, though. One is that the number of diagrams to consider is, a priori, much larger than in the off-shell case. The second, more dramatic one, is that the presence of light bridges makes both amplitudes, in the full model and in the EFT, non-local and, while their difference is guaranteed to be, it is very difficult to keep track of the delicate analytic cancellation between the non-localities in the two theories.

In this work we propose a numerical approach to on-shell matching that very efficiently side-steps this second disadvantage.[2] Our method can be used for any type of matching calculation, including tree-level or loop order finite matching or EFT renormalization and calculation of anomalous dimensions. It also prevents the need of using, and even defining, an evanescent basis of operators, as we will discuss below.[3] In complicated models, with many particles and couplings, the large number of diagrams to be computed makes on-shell matching less efficient than the standard off-shell one. Even in this case our approach can be used to reduce, in a fully automated way, any Green's basis to an arbitrary physical one or to translate between arbitrary physical bases. Existing computer tools can be then used to perform the off-shell matching to any Green's basis, and then our algorithm be used to perform the reduction (which, we remind the reader, needs to be done only once for each choice of Green's and physical bases).

Nonetheless, the very fact that on-shell matching relies on the computation of connected diagrams means that a certain amplitude with a large number of external particles will receive contributions from many different effective operators and can, therefore, be used to match many WCs at once. This is opposed to off-shell matching, in which the WCs of operators with different field content requires the calculation of different amplitudes. Thus, the larger number of diagrams is often compensated by the fact that a smaller number of amplitudes is needed to match all the WCs in the physical basis. The extra redundancy naturally appearing in off-shell matching, due to the freedom of using arbitrary off-shell external kinematics, is also present in our on-shell procedure. Indeed, since we use numerical kinematics, we can consider a larger number of on-shell kinematic configurations than strictly needed. The over constrained system of equations will have solution only if the calculation is correct. A final advantage of on-shell matching is that, since only physical observables are computed, gauge invariance is manifest

---

[1]Technically, the procedure is slightly different. In the diagrammatic approach the Green's basis is fixed implying that a complete Green's basis must be worked out from the beginning, but the reduction has to be done only once. In the functional approach the basis is not an input, but the resulting EFT Lagrangian must be reduced to a physical basis on a case-by-case analysis.

[2]See [41] for a similar approach to ours but with an analytic cancellation of non-local contributions that makes the procedure rather cumbersome and [42] for a recent proposal that uses amplitude methods to perform on-shell matching.

[3]This is due to our particular numerical approach that provides the matching directly in $d = 4$, rather than the fact that we perform the matching on- or off-shell. See Section 2.1 for further details.

even without the need of using the background field method, thus simplifying the construction of models.

We would like to close this section by emphasizing that the procedure we present in this article is entirely equivalent to other matching methods commonly used in the literature. The main difference is that our numerical on-shell matching approach is particularly efficient in certain circumstances, allowing calculations, like the reduction of some operators of the SMEFT dimension-8 Green's basis to the physical one, that had never been obtained before. It also provides a great deal of flexibility in the choice of physical basis.

The rest of this article is organized as follows. We describe the algorithm for doing on-shell matching in Section 2, making special emphasis on how to handle evanescent interactions. In Section 3, we provide examples of results obtained using our algorithm, including the reduction of Green's bases onto physical ones, the computation of beta functions as well the computation of finite matching contributions, including evanescent shifts. Some of these results extend previous knowledge in the literature. We conclude in Section 4, and leave some technical details for the appendices.

## 2 Numerical on-shell matching

The on-shell matching of a full model onto an EFT is performed by taking the difference of physical (connected, amputated, with external on-shell kinematics and the relevant factors of wave function renormalization constants) amplitudes in both theories. While the physical amplitudes, even after expanding in the heavy scales, are in general non-local, the difference is local, and can be absorbed in the WCs of local operators. Since we are matching physical amplitudes, we only need the WCs of the operators in a physical basis to match all the different contributions.

However, contrary to what happens in off-shell matching, in which the expansion in heavy scales together with the calculation of only the hard region contribution to loop integrals provides directly the local difference and no further calculation in the EFT is needed, here we need to compute the tree level EFT amplitude and cancel the non-local terms in the difference. Given the long intermediate expressions appearing in the calculation of the amplitudes, the analytic cancellation of the non-local terms becomes a formidable task. These non-localities arise from the presence of light bridges in the amplitudes and appear as poles in combinations of external momenta. Our proposal to avoid the explicit calculation of the cancellations is to compute the amplitudes for numerical values of the on-shell kinematics. Random kinematic configurations are generated using the algorithm developed in [43], combined with a custom algorithm based on [44]. This way the amplitudes are trivial to compute and the non-localities will cancel automatically. A failure of such cancellation, because of a mistake in the calculation or the lack of physical operators in the EFT will result in an inconsistent system of equations with no solution.

We perform the matching by subtracting renormalized physical amplitudes. We compute the physical amplitudes using dimensional regularization (dimReg) in $d = 4 - 2\epsilon$ space-time dimensions and renormalize them using the $\overline{\text{MS}}$ prescription. The matching is then performed, after taking the $\epsilon \to 0$ limit on the renormalized physical amplitudes, because numerical kinematics forces us to work in $d = 4$. In the off-shell matching, loop diagrams in the EFT correspond to the soft region contribution of loop diagrams in the full theory and therefore do not need to be computed. In our case, due to the fact that we are matching using physical, renormalized amplitudes in $d = 4$, the two might differ due to evanescent structures and we have to perform a partial calculation of both, as we discuss in the next section.

## 2.1 Evanescent operators

Evanescent operators are operators that vanish in $d = 4$ dimensions but are non-zero in general in $d = 4 - 2\epsilon$ dimensions. They are formally of rank $\epsilon$ and, when inserted in divergent loop diagrams, can multiply an ultraviolet (UV) $1/\epsilon$ pole and give a finite contribution (being a local effect it is not affected by infrared (IR) poles) that needs to be taken into account. Let us briefly describe a very simple example to fix ideas (a more detailed account is given below in Section 3.4 and the full discussion can be found in [45]). Consider the two operators

$$(\mathcal{O}_{\ell e})^{prst} = (\bar{\ell}^p \gamma^\mu \ell^r)(\bar{e}^s \gamma_\mu e^t), \tag{1}$$

$$(\mathcal{R}_{\ell e})^{prst} = (\bar{\ell}^p e^r)(\bar{e}^s \ell^t), \tag{2}$$

where $p, r, s, t$ are flavour indices and $\mathcal{O}_{\ell e}$ is in the Warsaw (physical) basis of the SMEFT whereas $\mathcal{R}_{\ell e}$ is not. The latter is however related to the former in 4 dimensions, thanks to Fierz identities

$$(\mathcal{R}_{\ell e})^{prst} \overset{d=4}{=} -\frac{1}{2}(\mathcal{O}_{\ell e})^{ptsr}. \tag{3}$$

Thus, we can define the evanescent operator

$$(\mathcal{E}_{\ell e})^{prst} \equiv (\mathcal{R}_{\ell e})^{prst} + \frac{1}{2}(\mathcal{O}_{\ell e})^{ptsr}. \tag{4}$$

In the simple example described in [45], a heavy copy of the SM Higgs doublet generates at tree level $\mathcal{R}_{\ell e}$. This can be reduced to $\mathcal{O}_{\ell e}$ but then the effect of $\mathcal{E}_{\ell e}$ must be included to account for the fact that the operator that was originally generated was $\mathcal{R}_{\ell e}$ rather than $\mathcal{O}_{\ell e}$. In summary, the correct result of the matching is, either the WC of $\mathcal{R}_{\ell e}$ and no evanescent shifts (which is not a result in the Warsaw basis), or the corresponding WC for the reduced operator $\mathcal{O}_{\ell e}$ accompanied by the relevant evanescent shifts, as reported in [45] and implemented in `MatchMakerEFT` [38].

This very procedure, the calculation of the finite contribution induced by the insertion of evanescent operators in one-loop diagrams hitting a UV $1/\epsilon$ pole can be also done in the on-shell version of matching (see the related discussion in [41]). However, our numerical procedure is done in 4 dimensions, after renormalisation, and we therefore obtain the matching in terms of $\mathcal{O}_{\ell e}$, with no history of whether $\mathcal{R}_{\ell e}$ or $\mathcal{O}_{\ell e}$ was originally generated in $d$ dimensions. Luckily there is a way to incorporate the evanescent effect in our procedure as follows. The authors of [46] emphasized that matching calculations could be simplified by using the method of regions [47]. Focusing on one-loop amplitudes we have, in the EFT,

$$\mathcal{M}_{\text{EFT}}^{(1)} = \mathcal{M}_{\text{EFT}}^{(1),\text{soft}} + \mathcal{M}_{\text{EFT}}^{(1),\text{hard}} = \mathcal{M}_{\text{EFT}}^{(1),\text{soft}}, \tag{5}$$

where we have used that, at one loop, there are only two relevant regions (soft, in which the loop momentum $k$ is of the same order as the light scales in the EFT; and hard, in which the loop momentum is much larger than all the other scales in the EFT) and the fact that the hard region contribution is scaleless in the EFT and therefore vanishes. Likewise, in the full theory we have

$$\mathcal{M}_{\text{full}}^{(1)} = \mathcal{M}_{\text{full}}^{(1),\text{soft}} + \mathcal{M}_{\text{full}}^{(1),\text{hard}}. \tag{6}$$

When the matching is performed in $d$ dimensions, the soft region contribution to the full theory is identical, by construction, to the one in the EFT; they cancel in the matching and only the hard region contribution to the full theory has to be computed. In the particular example we are discussing, both the EFT and the soft contribution to the full theory contain insertions of $\mathcal{R}_{\ell e}$ in $d$ dimensions and they are indeed identical.

In our approach to on-shell matching, however, we do the matching in 4 dimensions (after renormalization), and the EFT contains $\mathcal{O}_{\ell e}$ rather than $\mathcal{R}_{\ell e}$. The soft contribution of the full theory, being computed in $d$ dimensions, has instead insertions of $\mathcal{R}_{\ell e}$ and therefore, the two amplitudes are no longer equal. The finite part of the two amplitudes, arising from an $\mathcal{O}(\epsilon)$ term being multiplied by a UV $1/\epsilon$ pole gives precisely the evanescent shift that we are seeking. It should be emphasized that, following this procedure, we do not need to know which evanescent operators are generated, as the difference of the soft region expansions automatically produces the evanescent shift without the explicit construction of the evanescent operators. Of course that does not mean that we are not using a particular evanescent scheme. This is determined by our procedure and our treatment of $d$-dimensional structures. By using the naive dimensional regularization prescription for $\gamma^5$ (including a reading point prescription whenever needed) and the relevant basis of fermion bilinears, we automatically follow the scheme advocated in [45].

Note that we do not need to compute the full soft region contribution to the EFT and full theories but rather just the finite terms originated from UV poles. At one-loop order, these can be obtained upon expanding integrals in the region where the loop momentum $k$ is much greater that any external momentum or any light mass. Although this leads to scaleless integrals[4] that identically vanish in dimReg, as long as we separate IR and UV divergences we can correctly retrieve the UV poles. Indeed, this can be achieved by simply setting the integrals that scale like $1/k^4$ for large values of the loop momentum to $1/\epsilon_{\mathrm{UV}}$ (with the correct loop integral factors) and making every other integral to vanish. Schematically, the relevant part of the calculation reads, in either the EFT or the full theory,

$$\mathcal{M}^{(1),\mathrm{soft}}_{\mathrm{EFT/full}} = \frac{1}{\epsilon_{\mathrm{UV}}}[a_{\mathrm{EFT/full}} + b_{\mathrm{EFT/full}}\,\epsilon] + \ldots \xrightarrow{\text{renorm.}} b_{\mathrm{EFT/full}} + \ldots = \mathcal{M}^{(1),\mathrm{soft}}_{\mathrm{EFT/full}}\Big|_{\mathrm{UV}} + \ldots, \quad (7)$$

where the dots represent other finite terms that do not arise from UV poles and the arrow indicates renormalization. A few comments are in order. This expression is valid both for the EFT and the full theories. The divergent pieces, $a_{\mathrm{EFT/full}}/\epsilon_{\mathrm{UV}}$, can differ only due to terms proportional to the renormalization of evanescent operators.[5] The finite terms can also be different, the difference giving precisely the evanescent contribution we are looking for. We solely extract the pole from the loop integral, whether it is a scalar or a tensor one. This means that, when encountering a tensor integral, one should take special care in not keeping finite terms coming from the product of the $d = 4 - 2\epsilon$ factors explicitly appearing in the tensor reduction formula and the $1/\epsilon$ pole of the reduced scalar integral, since this is considered as a finite piece of the original loop integral. An extended discussion, with explicit examples, will be given in Section 3.4.

To summarize, the hard region contribution at one loop to the full theory plus the difference between the soft region contribution at one loop to the full and EFT theories is local, except for the non-localities induced by the light bridges, that are canceled by the equivalent ones in the EFT, and includes all the relevant evanescent shifts. It can therefore be matched by the contribution of local operators in the EFT.

## 2.2 Algorithmic on-shell matching

We are now in a position to provide an algorithmic procedure to efficiently perform on-shell matching. Explicit examples illustrating this algorithm will be discussed in Section 3.

---

[4]This is obvious in the EFT where no heavy scales remain. For the full theory, one must remember that we have previously expanded in the soft region, so that all heavy propagators have disappeared in favor of local terms and inverse powers of the heavy masses.

[5]We thank J. Fuentes-Martín for discussions on this issue.

1. Consider all the 1-particle-irreducible (1PI) contributions, up to the relevant order in operator dimension and loop order, to the 2-point functions. The location of the physical poles and the corresponding residues fix the on-shell conditions. The case of particle mixing can be treated perturbatively in the standard way.

2. Compute all the relevant connected, amputated amplitudes in both the full model and in the EFT, up to the correct order in the operator dimension. These amplitudes are to be computed in $d = 4 - 2\epsilon$ dimensions and renormalized with the $\overline{\text{MS}}$ prescription. After renormalization, we can set $\epsilon \to 0$ and work with four-dimensional renormalized amplitudes. The amplitudes that need to be computed in order to perform the matching up to the one-loop order are:

   - Tree-level amplitude in the EFT: $\mathcal{M}_{\text{EFT}}^{(0)}$.
   - One-loop finite part from UV poles in the EFT: $\mathcal{M}_{\text{EFT}}^{(1),\text{soft}}\Big|_{\text{UV}}$.
   - Tree-level amplitude in the full theory: $\mathcal{M}_{\text{full}}^{(0)}$.
   - One-loop hard region contribution in the full theory: $\mathcal{M}_{\text{full}}^{(1),\text{hard}}$.
   - One-loop finite part from UV poles of the soft region contribution in the full theory: $\mathcal{M}_{\text{full}}^{(1),\text{soft}}\Big|_{\text{UV}}$.

3. Once all the relevant renormalized amplitudes have been computed in $d = 4$, we consider first the amplitudes with the lowest number of external particles, to ensure that the system of equations to solve is always linear. For a fixed amplitude, we randomly generate as many on-shell kinematic configurations as WCs we need to solve for, replace them in the amplitudes and solve for the relevant WCs. As emphasized, we are working with renormalised physical amplitudes in 4 dimensions at this point and, therefore, we can replace numerically all the quantities appearing in the amplitude, including gamma matrices, fermion spinors or vector polarisations. We then proceed to other amplitudes with a larger number of external particles. In these amplitudes we replace the WCs that we have already solved for before attempting to solve the corresponding equations. The matching condition reads, at tree level,

$$\mathcal{M}_{\text{EFT}}^{(0)} = \mathcal{M}_{\text{full}}^{(0)}. \tag{8}$$

At one loop it includes the effect of possible evanescent structures, and reads

$$\mathcal{M}_{\text{EFT}}^{(0)} = \mathcal{M}_{\text{full}}^{(1),\text{hard}} + \mathcal{M}_{\text{full}}^{(1),\text{soft}}\Big|_{\text{UV}} - \mathcal{M}_{\text{EFT}}^{(1),\text{soft}}\Big|_{\text{UV}}. \tag{9}$$

Note that $\mathcal{M}_{\text{EFT}}^{(0)}$ is the tree-level amplitude in the EFT but now with one-loop sized WCs (to be determined from the matching condition). Also we have to replace the tree-level values of the WCs, that we determined from Eq. (8), in $\mathcal{M}_{\text{EFT}}^{(1),\text{soft}}\Big|_{\text{UV}}$, so that the right-hand side of the matching condition is completely known.

The algorithm we have just described leads to an efficient procedure for on-shell matching. It should be noted, however, that one can side-step the need of solving the amplitudes in growing number of external legs. Indeed, the fact that we need to consider connected amplitudes means that high-multiplicity amplitudes typically receive contribution from many WCs. We can use this property to solve for all these WCs with a single amplitude. The trick to avoid non-linear systems of equations in this case is to solve order by order in the loop expansion and mass dimension. This way we only encounter again linear systems of equations.

## 3 Specific examples

In this section we will provide a number of specific examples of our procedure and how it can be applied to the reduction of a Green's basis; the calculation of anomalous dimensions; or to finite matching, including evanescent shifts.

### 3.1 Reduction of the Green's basis of a $Z_2$-symmetric scalar up to mass dimension 8

Let us consider a Green's basis for a $Z_2$-symmetric scalar theory up to dimension 8. A suitable choice is given by

$$\mathcal{L} = \mathcal{L}_4 + \frac{1}{\Lambda^2}\mathcal{L}_6 + \frac{1}{\Lambda^4}\mathcal{L}_8 \,, \tag{10}$$

where

$$\mathcal{L}_4 = -\frac{1}{2}\phi(\partial^2 + m^2)\phi - \lambda\phi^4 \,, \tag{11}$$

$$\mathcal{L}_6 = \alpha_{61}\phi^6 + \beta_{61}\partial^2\phi\partial^2\phi + \beta_{62}\phi^3\partial^2\phi \,, \tag{12}$$

$$\mathcal{L}_8 = \alpha_{81}\phi^8 + \alpha_{82}\phi^2\partial_\mu\partial_\nu\phi\partial^\mu\partial^\nu\phi + \beta_{81}\phi\partial^6\phi$$
$$+ \beta_{82}\phi^3\partial^4\phi + \beta_{83}\phi^2\partial^2\phi\partial^2\phi + \beta_{84}\phi^5\partial^2\phi \,. \tag{13}$$

We have explicitly written the corresponding power of the cut-off $\Lambda$ to make the mass dimension apparent and have denoted with $\alpha_{di}$ and $\beta_{di}$ the WCs of physical and redundant operators of dimension $d$, respectively. In this example we will use on-shell matching at tree level to reduce the Green's basis onto the physical one. In order to do that we use the complete Green's basis as full model ($\mathcal{L}_{\text{Full}} = \mathcal{L}$) while the physical basis plays the role of the EFT ($\mathcal{L}_{\text{EFT}} = \mathcal{L}[\beta_{di} = 0]$).

Step 1 of our algorithm tells us to compute the 1PI contribution to the 2-point function. In the full theory it reads, at tree level and up to dimension 8,

$$\Pi(p^2) = -2p^4\frac{\beta_{61}}{\Lambda^2} + 2p^6\frac{\beta_{81}}{\Lambda^4} \,. \tag{14}$$

The standard procedure then fixes the physical mass,

$$p^2 - m^2 - \Pi(p^2)\Big|_{p^2 = m^2_{\text{phys}}} = 0 \quad \Rightarrow \quad m^2_{\text{phys}} = m^2\left[1 - 2\beta_{61}\frac{m^2}{\Lambda^2} + 2\left(\beta_{81} + 4\beta_{61}^2\right)\frac{m^4}{\Lambda^4}\right], \tag{15}$$

and the residue at the physical pole (a prime denotes derivative with respect to $p^2$)

$$Z = [1 - \Pi'(p^2 = m^2_{\text{phys}})]^{-1} = 1 - 4\beta_{61}\frac{m^2}{\Lambda^2} + 6(\beta_{81} + 4\beta_{61}^2)\frac{m^4}{\Lambda^4} \,. \tag{16}$$

The relation (15) can be inverted to express $m$ (the mass actually appearing in the calculations) in terms of the physical mass. In the EFT instead there are no 1PI corrections to the 2-point function so $m^2_{\text{phys}} = m^2$ and $Z = 1$. Note that the extra contributions to the 2-point function also modify the propagator appearing in light bridges which, in the full theory now reads

$$-\!-\!-\!-\!-\bigcirc-\!-\!-\!-\!- = \frac{i}{p^2 - m^2 - \Pi(p^2)}$$
$$= \frac{i}{p^2 - m^2}\left[1 - \frac{2}{\Lambda^2}\frac{p^4\beta_{61}}{p^2 - m^2} + \frac{2}{\Lambda^4}\frac{p^6(p^2 - m^2)\beta_{81} + 2p^8\beta_{61}^2}{(p^2 - m^2)^2} + \ldots\right]. \tag{17}$$

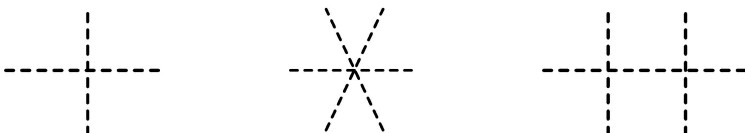

Figure 1: Tree-level topologies contributing to 4-scalar amplitudes (left) and 6-scalar amplitudes (center and right).

Once the physical mass, $Z$ factors and full propagators have been fixed, we compute the corresponding connected, amputated amplitudes, multiplying each external leg by a factor of $\sqrt{Z}$. After renormalizing the corresponding amplitudes we then set the on-shell condition for the external momenta $p^2 = m^2_{\text{phys}}$. Proceeding as mentioned in the algorithm from amplitudes with a smaller number of external particles to amplitudes with a larger number of particles we start with the 4-point amplitude, which receives contribution from a single topology, depicted in the left panel of Fig. 1. On the EFT side this amplitude receives contributions proportional to $\lambda$ and $\alpha_{82}$ whereas in the full theory it receives contributions, apart from these two couplings, from $\beta_{62}$, $\beta_{82}$ and $\beta_{83}$. Having two WCs in the EFT side we need to generate two independent on-shell kinematic configurations which allows us to solve for $\lambda$ and $\alpha_{82}$.

We next proceed to the 6-point amplitude, depicted in the center and right panels of Fig. 1. This amplitude receives a local 6-point contribution (center topology) proportional to $\alpha_{61}$ in the EFT and also to $\beta_{84}$ in the full theory, plus non-local terms proportional to two 4-point interactions (right topology). The latter involves WCs that have already been solved for on the EFT side and non-localities due to the light bridge that cancel between the EFT and the full theory. In this case a single on-shell kinematic configuration is enough to fix the value of $\alpha_{61}$.

Finally we move on to the 8-point amplitude, that receives contributions from topologies depicted in Fig. 2. The contributions split into a local one, proportional to $\alpha_{81}$, and non-local ones proportional to 4-point and 6-point vertices, which have already been solved for in the EFT side and whose non-local contributions cancel between the EFT and the full theory. Again a single on-shell kinematic configuration is enough to solve for $\alpha_{81}$.

Putting everything together, and writing back $m^2_{\text{phys}}$ in terms of $m^2$ on the full theory side, we obtain the correct reduction of the Green's basis onto the physical one, which reads

$$m^2 \rightarrow m^2 \left[ 1 - 2\beta_{61} \frac{m^2}{\Lambda^2} + 2(\beta_{81} + 4\beta_{61}^2) \frac{m^4}{\Lambda^4} \right], \tag{18}$$

$$\lambda \rightarrow \lambda + (\beta_{62} - 8\lambda\beta_{61}) \frac{m^2}{\Lambda^2} + \left( 64\lambda\beta_{61}^2 - 10\beta_{61}\beta_{62} + 12\lambda\beta_{81} - \beta_{82} - \beta_{83} \right) \frac{m^4}{\Lambda^4}, \tag{19}$$

$$\begin{aligned}
\alpha_{61} \rightarrow \alpha_{61} &+ 16\lambda^2\beta_{61} - 4\lambda\beta_{62} \\
&- \left( \frac{1728}{5}\lambda^2\beta_{61}^2 + \frac{22}{5}\beta_{62}^2 - \frac{512}{5}\lambda\beta_{61}\beta_{62} + 12\alpha_{61}\beta_{61} \right. \\
&\left. + \frac{304}{5}\lambda^2\beta_{81} - \frac{56}{5}\lambda\beta_{82} - 8\lambda\beta_{83} + \beta_{84} \right) \frac{m^2}{\Lambda^2},
\end{aligned} \tag{20}$$

$$\begin{aligned}
\alpha_{81} \rightarrow \alpha_{81} &- \frac{3072}{5}\lambda^3\beta_{61}^2 - \frac{108}{5}\lambda\beta_{62}^2 + \frac{1248}{5}\lambda^2\beta_{61}\beta_{62} - 48\lambda\alpha_{61}\beta_{61} + 6\alpha_{61}\beta_{62} \\
&- \frac{576}{5}\lambda^3\beta_{81} + \frac{144}{5}\lambda^2\beta_{82} + 16\lambda^2\beta_{83} - 4\lambda\beta_{84},
\end{aligned} \tag{21}$$

$$\alpha_{82} \rightarrow \alpha_{82}. \tag{22}$$



Figure 2: Tree-level topologies contributing to 8-scalar amplitudes.

Note that, as discussed above, we could perform the full matching using only the 8-point amplitude. As shown in Fig. 2, this amplitude receives contributions from operators with 4, 6 and 8 external legs and, provided we solve order by order in the mass dimension of the different contributions, we are left with linear systems of equations that provide the complete solution with this single amplitude. We describe a minimal but complete code in `Mathematica` to get the reduction to the physical basis in this way in the Appendix C. The full code is provided as an ancillary file in the `arXiv` submission of this manuscript.

These results can be also obtained via the reduction of the Green's basis using field redefinitions. Indeed, the following field redefinition removes the two dimension 6 redundant operators (up to dimension 8)

$$
\phi \to \phi \left( 1 - \frac{m^2 \beta_{61}}{\Lambda^2} + \frac{2m^4 \beta_{61}^2}{\Lambda^4} \right) + \partial^2 \phi \left( \frac{\beta_{61}}{\Lambda^2} - \frac{3m^2 \beta_{61}^2}{2\Lambda^4} \right) \\
+ \phi^3 \left( \frac{1}{\Lambda^2} (\beta_{62} - 4\lambda \beta_{61}) + \frac{2m^2 \beta_{61}}{\Lambda^4} (9\lambda \beta_{61} - 2\beta_{62}) \right).
\tag{23}
$$

The remaining dimension-8 operators can be eliminated via the equations of motion from Eq. (11), as non-linear terms are irrelevant at this order. Implementing carefully (and painfully) these replacements we have reproduced the results in Eqs. (18)-(22). We have also cross-checked these results with the help of `Matchete`.

## 3.2 Green's basis reduction in the SMEFT at dimension 8

The first phenomenologically useful application of our procedure is the reduction of redundant interactions of the SMEFT. A Green's basis for the SMEFT to dimension 6 was first worked out in [48]. This was later extended to dimension 8 in [49]. The relation between the redundant and physical operators to dimension 6 and below was also worked out in [48], while the bosonic dimension-8 redundant operators were related to physical terms in [50]. Here, as an example that goes beyond the state of the art, we provide the reduction of dimension-6 redundant operators up to dimension-8, which cannot be captured by equations of motion at linear order. (They can be accounted for by means of modified equations of motion [51].)

For clarity of the exposition, we limit ourselves to bosonic interactions with at most 6 fields, and work in the limit in which the only non-vanishing gauge coupling is $g_1$. We follow the conventions and notation in [48] and [28], using $c$ for the WCs of physical operators and $r$ for redundant ones. For ease of use, we collect the definition of all relevant operators in Appendix B. The corresponding reduction reads

$\boxed{H^2}$

$$
m_H^2 \to m_H^2 - m_H^4 r_{DH} + 2m_H^6 r_{DH}^2.
\tag{24}
$$

$\boxed{H^4}$

$$
\lambda \to \lambda - m_H^2 (4\lambda r_{DH} + 2r'_{HD}) + m_H^4 (16\lambda r_{DH}^2 + 10 r_{DH} r'_{HD}).
\tag{25}
$$

$\boxed{H^4 D^2}$

$$c_{H\Box} \rightarrow c_{H\Box} - \frac{1}{8} g_1^2 r_{2B} + \frac{1}{2} r'_{HD} + \frac{1}{2} g_1 r_{BDH} - m_H^2 (4 c_{H\Box} r_{DH} + g_1 r_{BDH} r_{DH} + 2 r_{DH} r'_{HD}), \quad (26)$$

$$c_{HD} \rightarrow c_{HD} - \frac{1}{2} g_1^2 r_{2B} + 2 g_1 r_{BDH} - m_H^2 (4 c_{HD} r_{DH} + 4 g_1 r_{BDH} r_{DH}). \quad (27)$$

$\boxed{H^6}$

$$\begin{aligned}
c_H \rightarrow c_H &+ \lambda^2 r_{DH} + \lambda r'_{HD} + m_H^2 \Big( \frac{1}{4} g_1^2 c_{HD} r_{2B} - \frac{1}{16} g_1^4 r_{2B}^2 - \frac{1}{2} g_1 c_{HD} r_{BDH} + \frac{1}{2} g_1^3 r_{2B} r_{BDH} \\
&- \frac{3}{4} g_1^2 r_{BDH}^2 - 6 c_H r_{DH} - \lambda c_{HD} r_{DH} + 8 \lambda c_{H\Box} r_{DH} + g_1 \lambda r_{BDH} r_{DH} - 11 \lambda^2 r_{DH}^2 \\
&- \frac{1}{2} c_{HD} r'_{HD} + 4 c_{H\Box} r'_{HD} + \frac{1}{2} g_1 r_{BDH} r'_{HD} - 9 \lambda r_{DH} r'_{HD} - \frac{1}{4} r'^2_{HD} - r''^2_{HD} \Big).
\end{aligned} \quad (28)$$

$\boxed{X^2 H^2}$

$$c_{HB} \rightarrow c_{HB} - 2 m_H^2 c_{HB} r_{DH}, \quad (29)$$

$$c_{H\widetilde{B}} \rightarrow c_{H\widetilde{B}} - 2 m_H^2 c_{H\widetilde{B}} r_{DH}. \quad (30)$$

$\boxed{H^4 D^4}$

$$c_{H^4}^{(1)} \rightarrow c_{H^4}^{(1)} + \frac{1}{2} g_1^2 r_{2B}^2 - 2 g_1 r_{2B} r_{BDH} + 2 r_{BDH}^2 + g_1^2 r_{B^2 D^4}, \quad (31)$$

$$c_{H^4}^{(2)} \rightarrow c_{H^4}^{(2)} - \frac{1}{2} g_1^2 r_{2B}^2 + 2 g_1 r_{2B} r_{BDH} - 2 r_{BDH}^2 - g_1^2 r_{B^2 D^4}. \quad (32)$$

$\boxed{X^2 H^4}$

$$\begin{aligned}
c_{B^2 H^4}^{(1)} \rightarrow &- c_{HB} g_1^2 r_{2B} + \frac{1}{16} g_1^4 r_{2B}^2 + 2 c_{HB} g_1 r_{BDH} - \frac{1}{4} g_1^3 r_{2B} r_{BDH} + \frac{1}{4} g_1^2 r_{BDH}^2 \\
&- 2 c_{HB} \lambda r_{DH} - c_{HB} r'_{HD},
\end{aligned} \quad (33)$$

$$c_{B^2 H^4}^{(2)} \rightarrow -g_1^2 c_{H\widetilde{B}} r_{2B} + 2 g_1 c_{H\widetilde{B}} r_{BDH} - 2 \lambda c_{H\widetilde{B}} r_{DH} - c_{H\widetilde{B}} r'_{HD}. \quad (34)$$

$\boxed{X H^4 D^2}$

$$c_{BH^4 D^2}^{(1)} \rightarrow c_{BH^4 D^2}^{(1)} - 4 g_1 c_{HB} r_{2B} + \frac{1}{2} g_1^3 r_{2B}^2 + 8 c_{HB} r_{BDH} - 2 g_1^2 r_{2B} r_{BDH} + 2 g_1 r_{BDH}^2, \quad (35)$$

$$c_{BH^4 D^2}^{(2)} \rightarrow c_{BH^4 D^2}^{(2)} - 4 g_1 c_{H\widetilde{B}} r_{2B} + 8 c_{H\widetilde{B}} r_{BDH}. \quad (36)$$

$\boxed{H^6 D^2}$

$$\begin{aligned}
c_{H^6}^{(1)} \rightarrow &- \frac{3}{4} g_1^2 c_{HD} r_{2B} + \frac{3}{16} g_1^4 r_{2B}^2 + \frac{3}{2} g_1 c_{HD} r_{BDH} - \frac{3}{2} g_1^3 r_{2B} r_{BDH} + \frac{9}{4} g_1^2 r_{BDH}^2 - \lambda c_{HD} r_{DH} \\
&- 8 \lambda c_{H\Box} r_{DH} - 3 g_1 \lambda r_{BDH} r_{DH} + \lambda^2 r_{DH}^2 - \frac{1}{2} c_{HD} r'_{HD} - 4 c_{H\Box} r'_{HD} - \frac{3}{2} g_1 r_{BDH} r'_{HD} \\
&- 3 \lambda r_{DH} r'_{HD} - \frac{7}{4} r'^2_{HD} + r''^2_{HD},
\end{aligned} \quad (37)$$

$$\begin{aligned}
c_{H^6}^{(2)} \rightarrow &- \frac{1}{2} g_1^2 c_{HD} r_{2B} + \frac{1}{8} g_1^4 r_{2B}^2 + g_1 c_{HD} r_{BDH} - g_1^3 r_{2B} r_{BDH} + \frac{3}{2} g_1^2 r_{BDH}^2 - 2 \lambda c_{HD} r_{DH} \\
&- 2 g_1 \lambda r_{BDH} r_{DH} - c_{HD} r'_{HD} - g_1 r_{BDH} r'_{HD}.
\end{aligned} \quad (38)$$

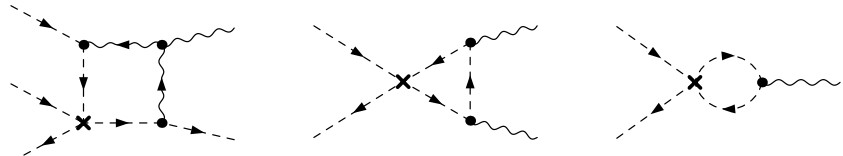

Figure 3: Representative diagrams contributing to the renormalization of $WH^4D^2$ (left), $W^2H^2D^2$ (center) and $WH^2D^4$ (right) by $H^4D^4$ operators (denoted by a cross). Insertions of renormalizable operators are denoted with a dot.

No other coefficient shifts under our assumptions. These results, extended to include both bosonic and fermionc terms, redundant and physical, as well as the full dependence on all SM couplings, have been utilized already in the renormalization of the two-fermion SMEFT interactions to dimension 8 [52], and can be found in the `Mathematica` notebook in https://github.com/SMEFT-Dimension8-RGEs.

### 3.3 Calculation of anomalous dimensions

One of the disadvantages of off-shell renormalization of EFTs is that divergences arising in 1PI off-shell Feynman diagrams cannot be captured by physical interactions only, but these must be extended with redundant operators. This problem is further strengthened in non-abelian gauge theories, since the Green's basis must include gauge-breaking interactions. One way to avoid this issue is using the background-field method [53], where gauge fields are split into classical backgrounds and gauge fluctuations. The gauge is fixed only for quantum fluctuations, which are in turn the only ones that enter in loops. Hence, the theory remains explicitly gauge invariant with respect to the background fields. However, this process also requires computing a much larger number of Feynman rules (since all particles can interact with both classical and quantum fields), which, at least within the most widely used tools for this purpose, in particular `FeynRules` [54], can become complicated for EFTs at high enough dimension.

Now, the aforementioned redundant and gauge-breaking terms obviously vanish on-shell, since physical quantities are gauge independent. Therefore, the computation of anomalous dimensions using our algorithm needs neither redundant operators nor the background field method. We can explicitly verify this using the example of the renormalization of the SMEFT dimension-8 operators in the class $WH^4D^2$ by loops involving $H^4D^4$ terms in the limit $g_1 \to 0$. We first note that, off-shell, $WH^4D^2$ operators receive contributions from redundant operators in other classes too, including $W^2H^2D^2$ and $WH^2D^4$. In particular [50]:

$$c_{WH^4D^2}^{(1)} \to c_{WH^4D^2}^{(1)} + 2r_{WH^4D^2}^{(7)} + g_2 r_{W^2H^2D^2}^{(11)} - 4g_2 r_{W^2H^2D^2}^{(19)} - \frac{1}{2}g_2^2 r_{WH^2D^4}^{(3)} + \dots, \tag{39}$$

$$c_{WH^4D^2}^{(2)} \to c_{WH^4D^2}^{(2)} - 4g_2 r_{W^2H^2D^2}^{(7)} + g_2 r_{W^2H^2D^2}^{(10)} + \dots, \tag{40}$$

$$c_{WH^4D^2}^{(3)} \to c_{WH^4D^2}^{(3)} + g_2 r_{W^2H^2D^2}^{(12)} + g_2 r_{W^2H^2D^2}^{(18)} + \dots, \tag{41}$$

while $c_{WH^4D^2}^{(4)}$ does not shift. The ellipses stand for terms that are not renormalized by $H^4D^4$. Accordingly, three different kind of 1PI diagrams, represented in Fig. 3, must be computed, and their divergences projected onto the physical basis. It was shown in [55] that they lead to (we denote with a tilde the WC that matches the corresponding divergence):

$$\tilde{c}_{WH^4D^2}^{(1)} = \frac{g_2}{96\pi^2}\left[(14\lambda + 19g_2^2)c_{H^4D^4}^{(1)} - 8(\lambda + 2g_2^2)c_{H^4D^4}^{(2)} - 18g_2^2 c_{H^4D^4}^{(3)}\right], \tag{42}$$

$$\tilde{r}_{WH^4D^2}^{(7)} = \frac{g_2}{16\pi^2}\left[(\lambda + 2g_2^2)c_{H^4D^4}^{(2)} - (\lambda + 2g_2^2)c_{H^4D^4}^{(1)}\right], \tag{43}$$

$$\tilde{r}_{WH^2D^4}^{(3)} = \frac{g_2}{192\pi^2}(c_{H^4D^4}^{(3)} - c_{H^4D^4}^{(2)}), \tag{44}$$

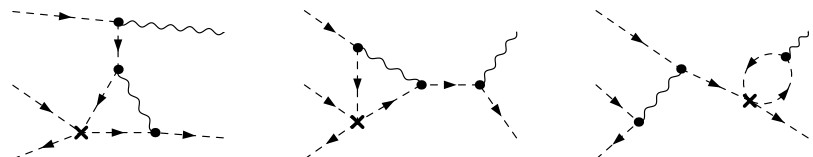

Figure 4: Representative diagams contributing to the on-shell renormalization of $WH^4D^2$ interactions. The cross denotes the insertion of an operator in the $H^4D^4$ class while the dots denote renormalizable couplings.

while all other divergences vanish.

Within the on-shell approach, however, we only need to compute the amplitude defined by $W^3(p_1)H^+(p_2)H^-(p_3)H^+(p_4)H^-(p_5)$ (which of course involves diagrams with light bridges like those represented in Fig. 4), extract the divergence, substitute the rational kinematics and project the result onto the physical basis. Effects from evanescent terms, that we discuss in detail in next section, can be neglected here, since one-loop anomalous dimensions can be entirely computed in $d = 4$ space-time dimensions, with the caveat that $\int d^4k/k^4 = i\pi^2/\epsilon$. We also avoid using the background-field method. Altogether, we obtain:

$$\tilde{c}_{WH^4D^2}^{(1)} = \frac{1}{384\pi^2} \left[ 4(2g_2\lambda - 5g_2^3)c_{H^4D^4}^{(1)} + (16g_2\lambda + 33g_2^3)c_{H^4D^4}^{(2)} - 73g_2^3 c_{H^4D^4}^{(3)} \right], \tag{45}$$

while other divergences in $WH^4D^2$ vanish. It can be trivially checked that Eqs. (39)–(41), together with Eqs. (42)–(44), agree with this result.

### 3.4 Finite matching and evanescent contributions

Let us now discuss finite one-loop matching, including evanescent shifts, in the context of on-shell matching. We will take as an example a model with a heavy copy of the SM Higgs boson and, for simplicity, we will assume that it only couples to the SM leptons,

$$\mathcal{L} = \mathcal{L}_{SM} + D_\mu\Phi^\dagger D^\mu\Phi - M^2\Phi^\dagger\Phi - \left(\mathcal{Y}^{pr}\bar{\ell}^p\Phi e^r + \text{h.c.}\right), \tag{46}$$

with $\mathcal{L}_{SM}$ the SM Lagrangian and we neglect other interactions allowed by the symmetries. This particular example was discussed, in the context of evanescent shifts, in [45].

The only operator in the Warsaw basis that is generated at tree level and mass dimension 6, with the couplings in Eq. (46), is $O_{\ell e}$, defined in Eq. (1), with coefficient

$$C_{\ell e}^{prst} = -\frac{1}{2M^2}\mathcal{Y}^{pt}(\mathcal{Y}^{rs})^*. \tag{47}$$

Indeed, computing at tree level the amplitude $\bar{e}_L^p(p_1)e_R^t(p_2)\bar{e}_R^s(p_3)e_L^r(p_4)$ with on-shell kinematics (we consider all incoming particles)

$$\sum_{i=1}^4 p_i = 0, \qquad p_i^2 = 0, \tag{48}$$

we obtain, in the UV and EFT theories, respectively,

$$\mathcal{M}_{UV} = \frac{\mathcal{Y}^{pt}(\mathcal{Y}^{rs})^*}{M^2} \bar{v}_1 P_R u_2 \bar{v}_3 P_L u_4, \tag{49}$$

$$\mathcal{M}_{EFT} = C_{\ell e}^{prst} \bar{v}_1 \gamma^\mu P_L u_4 \bar{v}_3 \gamma_\mu P_R u_2, \tag{50}$$

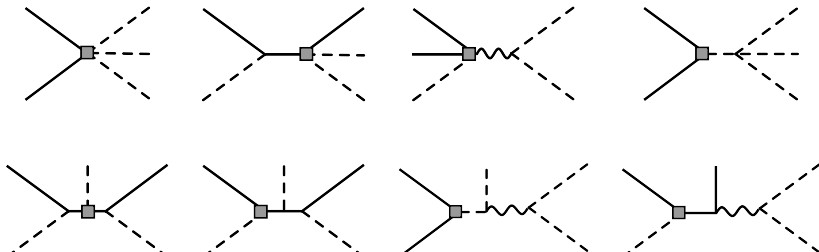

Figure 5: Tree-level topologies contributing to the physical amplitude of $\bar{\nu}_L e_R H^0 \bar{H}^0 H^-$ (all incoming) in the EFT involving one insertion of a one-loop sized WC (denoted by a square vertex).

where we have denoted the momentum with a subindex in the spinors and we have used the chirality projectors

$$P_{L,R} \equiv \frac{1 \mp \gamma^5}{2}. \tag{51}$$

Recall that, following our numerical procedure, we now have to replace (no renormalization needed at tree level) the spinors and the Dirac matrices by their numerical values with specific 4-d on-shell kinematic configurations. In $d = 4 - 2\epsilon$ dimensions the two fermionic structures are not related but in $d = 4$ the identity

$$\bar{v}_1 P_R u_2 \bar{v}_3 P_L u_4 = -\frac{1}{2} \bar{v}_1 \gamma^\mu P_L u_4 \bar{v}_3 \gamma_\mu P_R u_2 \qquad (d=4), \tag{52}$$

holds, from which, by equating the two amplitudes, we obtain the result in Eq. (47). In this particular case there are no relevant contributions with light bridges and, therefore, on-shell matching is essentially identical to off-shell one. However, as we have emphasized, our numerical procedure forces us to work in $d = 4$ dimensions. This means that, unless we look explicitly at the amplitudes before the numerical replacement, we would never know that, in $d \neq 4$ dimensions $\mathcal{R}_{\ell e}$ rather than $\mathcal{O}_{\ell e}$ is generated. Luckily enough, our procedure for evanescent contributions will, as we show below, take care of this fact automatically.

Let us now move on to the one-loop matching, for which the differences between on-shell and off-shell matching are more acute. Since we just want to exemplify these differences and the general procedure for finite (including evanescent shifts) matching, we will restrict ourselves to the operators that can be matched via the physical amplitude involving (all incoming particles) $\bar{\nu}_L^p, e_R^r, H^0, \bar{H}^0, H^-$. Such an amplitude receives contribution, at tree level and up to mass dimension 6, in the EFT from the following operators in the Warsaw basis

$$
\begin{aligned}
\mathcal{O}_\lambda &= -(H^\dagger H)^2, & \mathcal{O}_{He} &= (\bar{e}\gamma^\mu e)(H^\dagger \mathrm{i}\overset{\leftrightarrow}{D}_\mu H), \\
\mathcal{O}_{ye} &= -\bar{\ell}eH, & \mathcal{O}_{H\ell}^{(1)} &= (\bar{\ell}\gamma^\mu \ell)(H^\dagger \mathrm{i}\overset{\leftrightarrow}{D}_\mu H), \\
\mathcal{O}_{eB} &= (\bar{\ell}\sigma^{\mu\nu} e)HB_{\mu\nu}, & \mathcal{O}_{H\ell}^{(3)} &= (\bar{\ell}\gamma^\mu \sigma^I \ell)(H^\dagger \mathrm{i}\overset{\leftrightarrow}{D}{}^I_\mu H), \\
\mathcal{O}_{eW} &= (\bar{\ell}\sigma^{\mu\nu} e)\sigma^I H W^I_{\mu\nu}, & \mathcal{O}_{eH} &= (H^\dagger H)\bar{\ell}eH,
\end{aligned}
\tag{53}
$$

where the first two operators on the left column correspond to renormalizable ones (we have not written the kinetic terms that provide the corresponding gauge couplings) and the rest are dimension-6 operators. The different topologies contributing to this amplitude at tree level, in the EFT, with the insertion of a one-loop sized WC, are shown in Fig. 5. The fact that the amplitude receives contributions from all these operators means that we can match them all with the calculation of a single amplitude. This is a striking difference with respect to off-shell

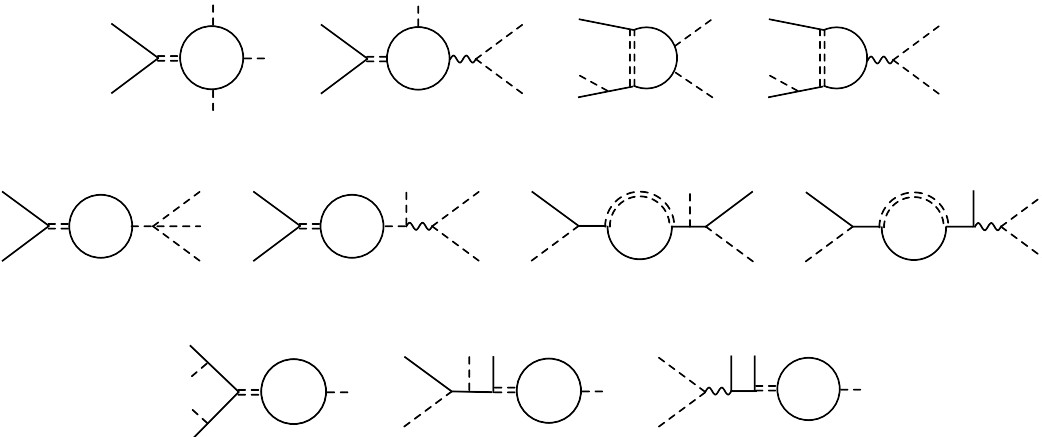

Figure 6: One loop topologies contributing to the physical amplitude of $\bar{\nu}_L e_R H^0 \bar{H}^0 H^-$ (all incoming) in the full theory involving, at least, one heavy propagator.

matching, for which only local contributions have to be considered in the EFT and therefore, this amplitude would only serve to match the operator $\mathcal{O}_{eH}$ (in the Green's basis), and can be used to reduce the number of amplitudes needed in the on-shell matching or as an extra cross-check of the calculation. Note that even the different gauge structures corresponding to $\mathcal{O}_{H\ell}^{(1,3)}$ can be disentangled with this amplitude, despite the fact that it has a "charge current structure", involving $\bar{\nu}_L e_R$. This is thanks to the top diagram in the second column of Fig. 5, in which the Higgs boson corresponding to the renormalizable coupling can be either neutral or charged, thus probing both gauge structures in the dimension-6 operator insertion.

The topologies corresponding to the contribution of the full theory to the amplitude are shown in Fig. 6. Note the presence of diagrams with light bridges, that would not be present in an off-shell matching, and also diagrams with only light particles circulating in the loop. The latter would also not be present in ordinary off-shell matching as they lead to scaleless integrals, that vanish in dimReg. For the same reason they do not contribute to the finite matching, given by the hard region contribution, in the on-shell approach. They are needed, however, in our case to provide the corresponding evanescent shifts that will arise from the soft region expansion. Also note that diagrams in which heavy particles circulate in the loop will, in general, contribute to both the finite matching (hard region) and evanescent shifts (soft region).

Finally, recall that, for the evanescent shifts, we need to compute some one-loop amplitudes in the EFT. These correspond to the same diagrams we show in Fig. 6, in which the heavy propagator is pinched into a local interaction, representing an insertion of the operator $\mathcal{O}_{\ell e}$. We do not display these diagrams explicitly, as they can be obtained directly from the ones in the figure.

Let us now describe in detail how to compute the different contributions to the matching conditions in Eq. (9). We start with the hard region contribution to the amplitude in the full theory, $\mathcal{M}_{\text{full}}^{(1),\text{hard}}$. This is computed as usual with the Taylor expansion in the region $k \sim M \gg p \sim m$, with $k$ the loop momentum, $p$ any combination of external momenta and $M, m$ the heavy and light masses, respectively,

$$\frac{1}{(k+p)^2 - M^2} = \frac{1}{k^2 - M^2} \sum_{n=0}^{\infty} \left( -\frac{p^2 + 2p \cdot k}{k^2 - M^2} \right)^n, \tag{54}$$

$$\frac{1}{(k+p)^2 - m^2} = \frac{1}{k^2} \sum_{n=0}^{\infty} \left( -\frac{p^2 + 2p \cdot k - m^2}{k^2} \right)^n. \tag{55}$$

Note that the first expression is equally valid for heavy bridges, just by setting $k = 0$. The order at which one should truncate the expansions is determined by the dimension of the EFT; up to dimension 6, terms proportional to $1/M^3$ or greater powers (after integration) should be discarded. In practice, this means that we should keep in the expansion only terms up to order $\mathcal{O}(1/k^6)$ as $k \to \infty$.

In the amplitude computation we consider only amputated diagrams,[6] but they have to be dressed with the corresponding power of the wave function renormalization constants, $\sqrt{Z}$, (defined from the residue of the physical pole in the two point function). The only $Z$ factors we need read[7]

$$[Z_\ell]^{pr} \equiv \delta^{pr} + [\delta Z_\ell]^{pr} = \delta^{pr} + \frac{1}{64\pi^2} \left(\mathcal{Y}\mathcal{Y}^\dagger\right)^{pr}, \tag{56}$$

$$[Z_e]^{pr} \equiv \delta^{pr} + [\delta Z_e]^{pr} = \delta^{pr} + \frac{1}{32\pi^2} \left(\mathcal{Y}^\dagger\mathcal{Y}\right)^{pr}, \tag{57}$$

$$Z_H = 0. \tag{58}$$

With this, the total hard region amplitude in the full theory reads, at one loop order,

$$\left[\mathcal{M}_{\text{full}}^{(1),\text{hard}}\right]^{pr} = \left[\mathcal{M}_{\text{full}}^{(1),\text{hard}}\right]^{pr}_{\substack{\text{amp} \\ \text{diags}}} + \frac{1}{2}[\delta Z_\ell]^{ps}\left[\mathcal{M}_{\text{full}}^{(0)}\right]^{sr} + \frac{1}{2}\left[\mathcal{M}_{\text{full}}^{(0)}\right]^{ps}[\delta Z_e]^{sr}, \tag{59}$$

where $p, r$ are the flavor indices of $\bar{\nu}_L$ and $e_R$, respectively. Matching this amplitude to the tree level in the EFT will automatically give us the finite matching directly in the Warsaw basis.

We also have to compute the correction to the physical masses, since the on-shell condition reads $p_i^2 = m_{i,\text{phys}}^2$, where $m_{i,\text{phys}}$ is the one-loop renormalized physical mass of particle $i$. In this case there is no shift to the physical mass due to the heavy particle and, therefore, $p^2 = 0$ for fermions and $p^2 = m_H^2$ for the Higgs.

Let us now move on to the calculation of the evanescent shifts. As described in detail in Section 2.1, we need to compute the finite contribution arising from UV poles times $\epsilon$ terms in the soft region contribution of the full and effective theories. In the full theory, the soft region contribution can be obtained by replacing the heavy propagators with their soft expansion,

$$\frac{1}{(k+p)^2 - M^2} = -\frac{1}{M^2}\sum_{n=0}^{\infty}\left(\frac{(k+p)^2}{M^2}\right)^n, \tag{60}$$

where the same expression can be used for heavy bridges upon setting $k = 0$. For the matching up to mass dimension 6 the series can be truncated at the first term, proportional to $1/M^2$.

Once the propagators have been expanded around the soft region, we need to isolate the UV poles of the remaining loop integrals. A straightforward way to achieve this, at one loop order, is by using again the expansion in Eq. (55) to extract the UV behavior of the amplitude. After this expansion all integrals are scaleless and vanish in dimReg. However, out of these scaleless integrals, the ones scaling as $1/k^4$, namely, integrals of the generic form $k^{\mu_1} \dots k^{\mu_n}/k^{n+4}$, contain the UV pole that has to be isolated. Once the UV pole of the loop integral has been found, all the relevant algebraic manipulations of the rest of the amplitude (including Dirac algebra) have to be performed in $d$ dimensions. The finite piece resulting from the product of the UV pole times the $\mathcal{O}(\epsilon)$ terms for these last manipulations is the one

---

[6]For practical reasons we do not amputate legs that generate mixing between the field $\Phi$ and any other SM field. In this model we can find such diagrams mixing $\Phi$ and $H$ (see, for instance, the last row of diagrams in Fig. 6). Including these diagrams is equivalent to perturbatively diagonalizing the corresponding one-loop mixing in the two point functions.

[7]Note that the first flavor index corresponds to the outgoing fermion. Also, since there is no tree-level contribution to this amplitude involving the heavy particle, we do not need the SM contribution to the wave function renormalization constant.

we are interested in. This procedure, computing the loop integral first, keeping only the UV pole, and then performing the remaining algebraic manipulations (in $d$ dimensions) has to be performed in this order. Otherwise the finite terms we obtain might not be the ones we are after. Let us explain this with the following simple example. Consider the amplitude

$$\mathcal{M} = \int \frac{\mathrm{d}^d k}{(2\pi)^d} \frac{\not{k}\not{k}}{k^6} = \gamma^\mu \gamma^\nu I_{\mu\nu}, \tag{61}$$

with the tensor integral

$$I_{\mu\nu} = \int \frac{\mathrm{d}^d k}{(2\pi)^d} \frac{k_\mu k_\nu}{k^6} = \frac{1}{d} \int \frac{\mathrm{d}^d k}{(2\pi)^d} \frac{g_{\mu\nu}}{k^4} = \frac{\mathrm{i} g_{\mu\nu}}{64\pi^2} \frac{1}{\epsilon_{\mathrm{UV}}} + \dots, \tag{62}$$

where the dots denote finite contributions that are not needed in the calculation. Thus, we have

$$\mathcal{M} = \gamma^\mu \gamma^\nu g_{\mu\nu} \frac{\mathrm{i}}{64\pi^2} \frac{1}{\epsilon_{\mathrm{UV}}} = \frac{\mathrm{i} d}{64\pi^2} \frac{1}{\epsilon_{\mathrm{UV}}} = \frac{\mathrm{i}}{16\pi^2} \frac{1}{\epsilon_{\mathrm{UV}}} - \frac{\mathrm{i}}{32\pi^2} + \dots \tag{63}$$

A naive use of the property $\not{k}\not{k} = k^2$ would have resulted in

$$\mathcal{M}\Big|_{\mathrm{naive}} = \frac{\mathrm{i}}{16\pi^2} \frac{1}{\epsilon_{\mathrm{UV}}}, \tag{64}$$

with no finite piece. Note the crucial fact that the $d$ in the denominator of Eq. (62) gets replaced with a 4, since we are only interested in the UV pole of the loop integral. Had we retained it as $d$, upon being inserted back in Eq. (61), it would have resulted in $\mathcal{M}|_{\mathrm{naive}}$ after working out the Dirac algebra. The reason behind is that keeping $d = 4 - 2\epsilon$ in Eq. (62) is equivalent to preserving order $\epsilon^0$ terms (after Taylor expanding) in the tensor loop integral $I_{\mu\nu}$.

The same procedure is used for the one-loop contributions in the EFT. In this case we do not need to explicitly perform the soft region expansion, as this corresponds to the full EFT amplitude. We just need to compute the UV pole from the tensor integral, as described in the preceding paragraph, and the finite contribution from the remaining terms in the amplitude. In this way, we can match the evanescent shift $\mathcal{M}_{\mathrm{full}}^{(1),\mathrm{soft}}\big|_{\mathrm{UV}} - \mathcal{M}_{\mathrm{EFT}}^{(1),\mathrm{soft}}\big|_{\mathrm{UV}}$ to the tree-level amplitude $\mathcal{M}_{\mathrm{EFT}}^{(0)}$.

Recall that we will eventually replace numerical kinematics in the amplitudes we have computed. This means that we can only match renormalized amplitudes. Inserting all our renormalized amplitudes in the matching condition, Eq. (9), and replacing with as many on-shell kinematic configurations as WCs we need to determine, we obtain a system of equations with the WCs as the only unknowns. This system can in general be non-linear but, solving it perturbatively in loop order and mass dimensions we always get a linear system.

After solving the system of equations from the matching condition we find the one-loop matching result in Eqs. (65)-(74), with the matching scale set to $\mu = M$. The evanescent shifts are explicitly shown in red. These results have been cross-checked using `Matchete` [37] and `MatchMakerEFT` [38] and the evanescent shifts reported in [45].

$$\lambda \to \lambda + \frac{g_2^4 m_H^2}{960\pi^2 M^2}, \tag{65}$$

$$y_e^{pr} \to y_e^{pr} - \frac{1}{128\pi^2} \left( \mathcal{Y}\mathcal{Y}^\dagger y_e + 2 y_e \mathcal{Y}^\dagger \mathcal{Y} \right)^{pr} + \frac{m_H^2}{32\pi^2 M^2} \mathcal{Y}^{pr} \left( \mathcal{Y}^\dagger \right)^{st} y_e^{ts}, \tag{66}$$

$$c_{HD} \to -\frac{g_1^4}{1920\pi^2 M^2}, \tag{67}$$

$$c_{H\Box} \to -\frac{1}{7680\pi^2 M^2}(g_1^4 + 3g_2^4), \tag{68}$$

$$c_{He}^{pr} \to \frac{g_1^4}{1920\pi^2 M^2}\delta^{pr} + \frac{7g_1^2}{576\pi^2 M^2}\left(\mathcal{Y}^\dagger \mathcal{Y}\right)^{pr} + \frac{1}{192\pi^2 M^2}\left(6\mathcal{Y}^\dagger y_e y_e^\dagger \mathcal{Y} + y^\dagger \mathcal{Y}\mathcal{Y}^\dagger y_e\right)^{pr}, \tag{69}$$

$$c_{Hl}^{(1)pr} \to \frac{g_1^4}{3840\pi^2 M^2}\delta^{pr} + \frac{17g_1^2}{1152\pi^2 M^2}\left(\mathcal{Y}\mathcal{Y}^\dagger\right)^{pr} - \frac{1}{192\pi^2 M^2}\left(6\mathcal{Y}y_e^\dagger y_e \mathcal{Y}^\dagger + y_e \mathcal{Y}^\dagger \mathcal{Y} y_e^\dagger\right)^{pr}, \tag{70}$$

$$c_{Hl}^{(3)pr} \to -\frac{g_2^4}{3840\pi^2 M^2}\delta^{pr} + \frac{g_2^2}{1152\pi^2 M^2}\left(\mathcal{Y}\mathcal{Y}^\dagger\right)^{pr} - \frac{1}{192\pi^2 M^2}\left(y_e \mathcal{Y}^\dagger \mathcal{Y} y_e^\dagger\right)^{pr}, \tag{71}$$

$$c_{eB}^{pr} \to -\frac{g_1}{768\pi^2 M^2}\left(5\mathcal{Y}\mathcal{Y}^\dagger y_e + 2y_e \mathcal{Y}^\dagger \mathcal{Y}\right)^{pr} + \frac{3g_1}{128\pi^2 M^2}\mathcal{Y}^{pr}\left(\mathcal{Y}^\dagger\right)^{st} y_e^{ts}, \tag{72}$$

$$c_{eW}^{pr} \to -\frac{5g_2}{768\pi^2 M^2}\left(\mathcal{Y}\mathcal{Y}^\dagger y_e\right)^{pr} - \frac{g_2}{128\pi^2 M^2}\mathcal{Y}^{pr}\left(\mathcal{Y}^\dagger\right)^{st} y_e^{ts}, \tag{73}$$

$$c_{eH}^{pr} \to -\frac{g_2^4}{3840\pi^2 M^2}y_e^{pr} + \frac{1}{192\pi^2 M^2}\left(y_e y_e^\dagger \mathcal{Y}\mathcal{Y}^\dagger y_e + 2y_e \mathcal{Y}^\dagger \mathcal{Y} y^\dagger y - 3\mathcal{Y}y_e^\dagger y_e \mathcal{Y}^\dagger y_e \tag{74}$$

$$-3y_e \mathcal{Y}^\dagger y_e y_e^\dagger \mathcal{Y}\right)^{pr} + \frac{1}{16\pi^2 M^2}\mathcal{Y}^{pr} y_e^{st}\left(y_e^\dagger\right)^{tu} y_e^{uv}\left(\mathcal{Y}^\dagger\right)^{vs} - \frac{\lambda}{32\pi^2 M^2}\mathcal{Y}^{pr}\left(\mathcal{Y}^\dagger\right)^{st} y_e^{ts}.$$

## 4 Conclusions and outlook

Effective field theories have become an essential part of every theorist's toolbox. Off-shell matching is now fully automated, both in the diagrammatic and functional approaches, but it requires the reduction of a Green's basis into the physical one. This reduction, while straightforward in principle, is tedious, error prone and not easy to fully automate, in particular when going to higher orders in the power counting expansion. Furthermore, it is highly non-trivial to devise a generic reduction algorithm that allows to reduce the result to a user-chosen physical basis.

On-shell matching, much less developed and barely adopted by the community, is an alternative approach to matching that only requires the use of a physical basis. No redundant or evanescent operators have to be included in the basis and no reduction is needed. The main drawback, besides the technical one of having to consider a larger number of diagrams, is the fact that the amplitudes in the full and effective theories are both non local and these non localities only cancel in the difference. This cancellation is very difficult to perform analytically, given the large number of terms involved, what becomes the main obstacle to successfully perform the matching on-shell.

In this work we have proposed an efficient algorithm to perform on-shell matching that side-steps this obstacle by numerically evaluating the physical amplitudes. This is done by generating random numerical on-shell kinematics, in the field of rational numbers, and evaluating the corresponding physical amplitudes, after renormalization, for these kinematic configurations. In this way we obtain a set of linear system of equations with the WCs we want to compute as the only unknowns. The use of rational numbers of the kinematic configurations ensures an analytic solution with no round-off errors. Our numerical procedure is very efficient in providing the required cancellation of non-local terms in the difference between the full theory and the EFT amplitudes but it forces us to work with renormalized amplitudes in $d = 4$ dimensions. This requires a careful treatment of the soft contribution to the amplitude in the full and effective theories, as they might be different due to evanescent effects. We have taken advantage of this property to automatically include the calculation of evanescent shifts in our algorithm. Thus, following this algorithm, we obtain, in an efficient and straight-forward way, the contribution to the finite matching, including evanescent shifts whenever relevant, directly in the physical basis. Incidentally, given that we use physical observables to perform

the matching, we do not need to use the background field gauge to match all the contributions with gauge invariant operators.

As practical examples we have shown how to use our algorithm in several different situations, including: the reduction of a Green's basis to a physical one; the calculation of the $\beta$ functions of an EFT, directly in the physical basis; or the calculation of a finite one-loop matching, including evanescent shifts, in a model with heavy particles.

Our efficient on-shell procedure has been implemented in a `Mathematica` package, that will be documented elsewhere [56] and, in future releases, we expect it to become a standard feature of `MatchMakerEFT` [38].

# Acknowledgments

We gratefully acknowledge very useful discussions with S. de Angelis, J.C. Criado, R. Fonseca, J. Fuentes-Martín, A. Lazopoulos, P. Olgoso, M. Pérez-Victoria and R. Pittau.

**Funding information** This work has been partially supported by MCIN/AEI (10.13039/501100011033) and ERDF (grants PID2021-128396NB-I00, PID2022-139466NB-C21 and PID2022-139466NB-C22), by the Junta de Andalucía grants FQM 101 and P21-00199 and by Consejería de Universidad, Investigación e Innovación, Gobierno de España and Unión Europea – NextGenerationEU under grants AST22_6.5 and CNS2022-136024. MC is further supported by the Ramón y Cajal program RYC2019-027155-I.

# A  Rational on-shell kinematics

The use of numerical methods to solve systems of equations can face the problem of numerical accuracy. In order to avoid this problem, we generate on-shell numerical kinematics in the field of rational numbers. For this purpose, we use the algorithm developed in [43] alongside a custom algorithm based on [44]. The advantage of this method, besides the fact that rational kinematics guarantees exactness of the solution, is that we can even enforce independent symbolic masses for each particle in the on-shell condition. This is crucial to obtain the (light) mass dependence on the matching procedure.

The spinor-helicity formalism maps the four-momentum space into the realm of complex $2\times2$ matrices, allowing any vector to be expressed as the product of two Weyl spinors. Given a left-handed spinor, $\lambda_\alpha$, and a right-handed spinor, $\widetilde{\lambda}^{\dot\alpha}$, the four-momentum vector $p^\nu$ associated to a massless particle is given by

$$p^\nu = \frac{1}{2}\lambda^\alpha \widetilde{\lambda}^{\dot\alpha} \sigma^\nu_{\alpha\dot\alpha}, \tag{A.1}$$

where $\lambda^\alpha \equiv \epsilon^{\alpha\beta}\lambda_\beta$, with $\epsilon^{\alpha\beta}$ the totally antisymmetric tensor with $\epsilon^{12} = 1$, and $\sigma^\nu = (1_{2\times2}, \sigma^I)$, with $\sigma^I$ the Pauli matrices.

For processes involving particles of spin-1, we will require polarization vectors. In terms of spinors, we can write

$$\varepsilon^\nu_+ = \frac{1}{\sqrt{2}}\frac{\lambda^\alpha \widetilde{\mu}^{\dot\alpha}\sigma^\nu_{\alpha\dot\alpha}}{\lambda_\beta \mu^\beta}, \quad \text{and} \quad \varepsilon^\nu_- = \frac{1}{\sqrt{2}}\frac{\mu^\alpha \widetilde{\lambda}^{\dot\alpha}\sigma^\nu_{\alpha\dot\alpha}}{\widetilde{\lambda}_{\dot\beta}\widetilde{\mu}^{\dot\beta}}, \tag{A.2}$$

where $\mu$ and $\widetilde{\mu}$ are auxiliary spinors, commonly referred to as reference spinors and $\widetilde{\lambda}_{\dot\alpha} \equiv \epsilon_{\dot\alpha\dot\beta}\widetilde{\lambda}^{\dot\beta}$, with $\epsilon_{\dot1\dot2} = -1$. These definitions satisfy $p_\nu \varepsilon^\nu_\pm = 0$ and $\varepsilon_{+\nu}\varepsilon^\nu_- = -1$, which are the properties required for physical polarizations.

For spin-$\frac{1}{2}$ particles, spinors satisfying the Dirac equation are required. Here, we consider incoming massless fermions, meaning that the associated Dirac spinors $u$ and $\bar{v}$ must satisfy

$$\not{p}u = 0, \quad \text{and} \quad \bar{v}\not{p} = 0. \tag{A.3}$$

The connection between left- and right-handed Dirac spinors and the Weyl spinor basis is established by the relations

$$P_L u(p) = \begin{pmatrix} \widetilde{\lambda}_{\dot{\alpha}} \\ 0 \end{pmatrix}, \qquad P_R u(p) = \begin{pmatrix} 0 \\ \lambda^{\alpha} \end{pmatrix}, \qquad \bar{v}(p)P_R = \begin{pmatrix} \widetilde{\lambda}^{\dot{\alpha}} \\ 0 \end{pmatrix}, \quad \text{and} \quad \bar{v}(p)P_L = \begin{pmatrix} 0 \\ \lambda_{\alpha} \end{pmatrix}, \tag{A.4}$$

where $\lambda$ and $\widetilde{\lambda}$ represent the spinors associated with the four-momentum $p$.

For the massive case, we can write the four-momentum in terms of two pairs of spinors $(\lambda, \tilde{\lambda}), (\mu, \tilde{\mu})$ as

$$p^{\nu} = \frac{1}{2}\left( \lambda^{\alpha}\widetilde{\lambda}^{\dot{\alpha}} + \frac{m^2}{\mu^{\beta}\widetilde{\mu}^{\dot{\beta}}\lambda_{\beta}\widetilde{\lambda}_{\dot{\beta}}}\mu^{\alpha}\widetilde{\mu}^{\dot{\alpha}} \right)\sigma_{\alpha\dot{\alpha}}^{\nu}, \tag{A.5}$$

with $p^2 = m^2$.

# B SMEFT conventions and operators

The conventions for the leptonic EW sector of the SM Lagrangian used throughout this work are the following:

$$\mathcal{L}_{\text{SM}} = -\frac{1}{4}W_{\mu\nu}^{I}W^{I\mu\nu} - \frac{1}{4}B_{\mu\nu}^{I}B^{I\mu\nu} + \left(D_{\mu}H\right)^{\dagger}\left(D^{\mu}H\right) - m_H^2\left(H^{\dagger}H\right)$$
$$- \frac{\lambda}{2}\left(H^{\dagger}H\right)^2 + \bar{\ell}\,\mathrm{i}\not{D}\ell + \bar{e}\,\mathrm{i}\not{D}e - \left(y_e^{pr}\bar{\ell}^p H e^r + \text{h.c.}\right), \tag{B.1}$$

with the corresponding gauge-fixing and ghost Lagrangians. The covariant derivative is

$$D_{\mu} = \partial_{\mu} - \frac{\mathrm{i}}{2}g_2\sigma^I W_{\mu}^I - \mathrm{i}g_1 Y B_{\mu}, \tag{B.2}$$

where $\sigma^I$ are the Pauli matrices, $Y$ the hypercharge and $g_1, g_2$ the $U(1)$ and $SU(2)$ gauge couplings, respectively.

In tables 1 and 2 we describe the notation for the dimension-6 and dimension-8 bosonic operators in the SMEFT.

# C Sample code for the tree-level reduction of Green's basis

In this Appendix we will show and explain the Mathematica code to find the reduction of the scalar example in Sec. 3.1 up to dimension 8. We will make use of the package combination FeynCalc+FeynArts [58, 59] which can be loaded into the Mathematica interface via the commands:

```
In[1]:= $LoadAddOns = {"FeynArts"};
        << FeynCalc`
```

Note that this initialization procedure only works correctly if FeynCalc is in the path. Two models will be loaded, namely, the Lagrangian with the full basis up to dimension 8 and the Lagrangian in the physical basis, that is, with every coefficient $\beta_{di}$ set to zero. (see

Table 1: Operators of the dimension 6 Green's basis in the bosonic sector of the SMEFT containing only the gauge boson $B$ [48]. The redundant operators are the gray ones.

| Dimension 6 Bosonic Sector | | | | |
|---|---|---|---|---|
| $H^6$ | $\mathcal{O}_H$ | $(H^\dagger H)^3$ | | |
| $H^4 D^2$ | $\mathcal{O}_{H\square}$ | $(H^\dagger H)\square(H^\dagger H)$ | $\mathcal{O}_{HD}$ | $(H^\dagger D^\mu H)^\dagger(H^\dagger D_\mu H)$ |
| | $\mathcal{O}'_{HD}$ | $(H^\dagger H)(D_\mu H)^\dagger(D^\mu H)$ | $\mathcal{O}''_{HD}$ | $i(H^\dagger H)D_\mu(H^\dagger D^\mu H - D^\mu H^\dagger H)$ |
| $H^2 D^4$ | $\mathcal{O}_{DH}$ | $(D_\mu D^\mu H)^\dagger(D_\nu D^\nu H)$ | | |
| $X^2 H^2$ | $\mathcal{O}_{HB}$ | $B_{\mu\nu}B^{\mu\nu}(H^\dagger H)$ | $\mathcal{O}_{H\widetilde{B}}$ | $\widetilde{B}_{\mu\nu}B^{\mu\nu}(H^\dagger H)$ |
| $H^2 X D^2$ | $\mathcal{O}_{BDH}$ | $\partial_\nu B^{\mu\nu}(H^\dagger i\overleftrightarrow{D}_\mu H)$ | | |
| $X^2 D^2$ | $\mathcal{O}_{2B}$ | $-\frac{1}{2}(\partial_\mu B^{\mu\nu}\partial^\rho B_{\rho\nu})$ | | |

Table 2: Operators of the dimension 8 Green's basis in the bosonic sector of the SMEFT appearing in the text [50], [57]. The redundant operators are the gray ones.

| Dimension 8 Bosonic Sector | | | | |
|---|---|---|---|---|
| $H^6 D^2$ | $\mathcal{O}^{(1)}_{H^6}$ | $(H^\dagger H)^2(D_\mu H^\dagger D^\mu H)$ | $\mathcal{O}^{(2)}_{H^6}$ | $(H^\dagger H)(H^\dagger\sigma^I H)(D_\mu H^\dagger\sigma^I D^\mu H)$ |
| $H^4 D^4$ | $\mathcal{O}^{(1)}_{H^4}$ | $(D_\mu H^\dagger D_\nu H)(D^\nu H^\dagger D^\mu H)$ | $\mathcal{O}^{(2)}_{H^4}$ | $(D_\mu H^\dagger D_\nu H)(D^\mu H^\dagger D^\nu H)$ |
| | $\mathcal{O}^{(3)}_{H^4}$ | $(D^\mu H^\dagger D_\mu H)(D^\nu H^\dagger D_\nu H)$ | | |
| $X^2 H^4$ | $\mathcal{O}^{(1)}_{B^2 H^4}$ | $(H^\dagger H)^2 B_{\mu\nu}B^{\mu\nu}$ | $\mathcal{O}^{(2)}_{B^2 H^4}$ | $(H^\dagger H)^2\widetilde{B}_{\mu\nu}B^{\mu\nu}$ |
| $XH^4 D^2$ | $\mathcal{O}^{(1)}_{BH^4 D^2}$ | $i(H^\dagger H)(D^\mu H^\dagger D^\nu H)B_{\mu\nu}$ | $\mathcal{O}^{(2)}_{BH^4 D^2}$ | $i(H^\dagger H)(D^\mu H^\dagger D^\nu H)\widetilde{B}_{\mu\nu}$ |
| | $\mathcal{O}^{(1)}_{WH^4 D^2}$ | $i(H^\dagger H)(D^\mu H^\dagger\sigma^I D^\nu H)W^I_{\mu\nu}$ | $\mathcal{O}^{(2)}_{WH^4 D^2}$ | $i(H^\dagger H)(D^\mu H^\dagger\sigma^I D^\nu H)\widetilde{W}^I_{\mu\nu}$ |
| | $\mathcal{O}^{(3)}_{WH^4 D^2}$ | $i\epsilon^{IJK}(H^\dagger\sigma^I H)(D^\mu H^\dagger\sigma^J D^\nu H)W^K_{\mu\nu}$ | $\mathcal{O}^{(4)}_{WH^4 D^2}$ | $i\epsilon^{IJK}(H^\dagger\sigma^I H)(D^\mu H^\dagger\sigma^J D^\nu H)\widetilde{W}^K_{\mu\nu}$ |
| $X^2 H^2 D^2$ | $\mathcal{O}^{(7)}_{W^2 H^2 D^2}$ | $i\epsilon^{IJK}(H^\dagger\sigma^I D^\nu H - D^\nu H^\dagger\sigma^I H)D_\mu W^{J\mu\rho}\widetilde{W}^K_{\nu\rho}$ | $\mathcal{O}^{(10)}_{W^2 H^2 D^2}$ | $(H^\dagger D_\nu H + D_\nu H^\dagger H)D_\mu W^{I\mu\rho}\widetilde{W}^I_\rho{}^\nu$ |
| | $\mathcal{O}^{(11)}_{W^2 H^2 D^2}$ | $(H^\dagger D_\nu H + D_\nu H^\dagger H)D_\mu W^{I\mu\rho}W^I_\rho{}^\nu$ | $\mathcal{O}^{(12)}_{W^2 H^2 D^2}$ | $i(H^\dagger D_\nu H - D_\nu H^\dagger H)D_\mu W^{I\mu\rho}W^I_\rho{}^\nu$ |
| | $\mathcal{O}^{(18)}_{W^2 H^2 D^2}$ | $\epsilon^{IJK}(H^\dagger\sigma^I D^\nu H + D^\nu H^\dagger\sigma^I H)D_\mu W^{J\mu\rho}W^K_{\nu\rho}$ | $\mathcal{O}^{(19)}_{W^2 H^2 D^2}$ | $i\epsilon^{IJK}(H^\dagger\sigma^I D^\nu H - D^\nu H^\dagger\sigma^I H)D_\mu W^{J\mu\rho}W^K_{\nu\rho}$ |
| $XH^2 D^4$ | $\mathcal{O}^{(3)}_{WH^2 D^4}$ | $i(D_\rho D_\nu H^\dagger\sigma^I D^\rho H - D^\rho H^\dagger\sigma^I D_\rho D_\nu H)D_\mu W^{I\mu\nu}$ | | |

Eqs. (10)-(13)). For practical purposes, in the present code we represent $\alpha_{di}$ (physical) and $\beta_{di}$ (redundant) coefficients as adi and rdi, respectively. The .fr files from which the FeynArts models can be created via FeynRules [54] can be found as ancillary files.

Before entering into the details of the code, we also define a function to expand any expression exp to order $n$ in the perturbative parameter $1/\Lambda^2$ (in the code invL2), which will be very useful throughout the discussion.

```
In[2]:= EFTSeries[exp_, n_] := Normal@Series[exp, {invL2, 0, n}]
```

This way, we will be keeping only terms up to dimension 8 (this is, $1/\Lambda^4$) by setting $n = 2$.

The code can be divided in three main sections. The first one is dedicated to obtain, from the 2-point function, the physical mass and the residue in the pole $Z$. The 2-point function needs to be extracted with other tools (e.g., from a notebook with FeynRules loaded, upon setting FR$Loop = True and computing the Feynman rules for the Lagrangian). From there, we can extract the $\Pi(p^2)$ function, which in the full model reads

```
In[3]:= pi[p2_] := -2 invL2 r61 p2^2 + 2 invL2^2 r81 p2^3
```

To extract the physical pole (i.e., the physical mass), we expand it in its different EFT orders and perturbatively compute $m^2_{\text{phys}} - m^2_0 - \Pi(m^2_{\text{phys}}) = 0$, with $m_0$ the mass term in the Lagrangian of the full model. This is saved into the mphysCondition variable as a list, containing the different expressions obtained order by order in invL2.

```
In[4]:= mphys2 = mphys2dim4 + invL2 mphys2dim6 + invL2^2 mphys2dim8;
```

```
In[5]:= mphysCondition =
        CoefficientList[EFTSeries[mphys2 - m0^2 - pi[mphys2], 2], invL2];
```

The only thing left to do is to compute the values for mphys2dim4, mphys2dim6 and mphys2dim8 by setting every element of mphysCondition to zero and, then, compute the residue $Z$.

```
In[6]:= mphys2 = mphys2 /. Flatten@
        Solve[mphysCondition == 0, {mphys2dim4, mphys2dim6, mphys2dim8}]
        Z = EFTSeries[(1 - pi'[mphys2])^(-1), 2]
```

```
Out[6]= m0^2 - 2 invL2 m0^4 r61 + 2 invL2^2 (4 m0^6 r61^2 + m0^6 r81)
```

```
Out[7]= 1 - 4 invL2 m0^2 r61 + invL2^2 (24 m0^4 r61^2 + 6 m0^4 r81)
```

This has been performed for the full model and the same should be repeated for the physical one. However, since we do not add any extra 2-point operator to this model apart from the kinetic and mass terms, the physical mass is directly the mass in the Lagrangian (which is, in turn, identical to the physical mass that has just been computed) and the residue is $Z = 1$. We finally set two replacement rules to change propagators (which FeynCalc expresses as FeynAmpDenominator structures) to their expression depending on whether the amplitude is computed with the full or the physical model.

```
In[8]:= PropFull[p_] := EFTSeries[1/(SP[p, p] - m0^2 - Pi[SP[p, p]]), 2]
        PropPhys[p_] := EFTSeries[1/(SP[p, p] - mphys2), 2]
        expandPropFull = {FeynAmpDenominator[PD[Momentum[p_], m_]] :> PropFull[p]};
        expandPropPhys = {FeynAmpDenominator[PD[Momentum[p_], m_]] :> PropPhys[p]};
```

To facilitate the expansion of various expressions in the perturbative parameter, we introduce a replacement rule that associates each coefficient with the corresponding power of invL2. Additionally, since we will solve order by order in the mass dimension of the different contributions, it is necessary to explicitly expand the coefficients as a series in powers of invL2.

```
In[9]:= EFTOrder = Flatten@
          {(# -> invL2 #) &/@ {a61, r61, r62, a61phys},
           (# -> invL2^2 #) &/@ {a81, a82, r81, r82, r83, r84, a81phys, a82phys}};
        perturbativeOrder = {
            lmbdphys -> lmbd + invL2 lmbdphysdim6 + invL2^2 lmbdphysdim8,
            a61phys -> a61physdim6 + invL2 a61physdim8,
            a81phys -> a81physdim8,
            a82phys -> a82physdim8
          };
```

In this example, we perform the full matching using only the 8-point amplitude. Specifically we consider a process with eight incoming scalars $S[1] \equiv \phi$. For convenience, we also define a list representing the momenta of the external legs. We then compute the amplitudes with both Lagrangians, multiplying by the $Z$ factor the amplitude calculated within the full basis (`ampFull`) and substituting the corresponding propagators in each case. For the amplitude in terms of the physical basis (`ampPhys`), we use the `perturbativeOrder` replacement rule too so that we can solve for the coefficients perturbatively in the EFT order.

```
In[10]:= nScalars = 8;
        process = Table[S[1], {i, nScalars}] -> {};
        momenta = Table[Symbol["P" <> ToString[i]], {i, nScalars}];

In[11]:= topo = CreateTopologies[0, nScalars -> 0, Adjacencies -> {4, 6, 8}];

In[12]:= diagsFull = InsertFields[topo, process, InsertionLevel -> {Particles},
            Model -> modelFull, GenericModel -> modelFull];
        ampFull = FCFAConvert[CreateFeynAmp[diagsFull],
            IncomingMomenta -> momenta, List -> False] /. EFTOrder;

In[13]:= ampFull =
            EFTSeries[Z^(nScalars/2), 2] EFTSeries[ampFull, 2] /. expandPropFull;
        ampFull = EFTSeries[ampFull, 2] // ExpandScalarProduct;

In[14]:= diagsPhys = InsertFields[topo, process, InsertionLevel -> {Particles},
            Model -> modelPhys, GenericModel -> modelPhys];
        ampPhys = FCFAConvert[CreateFeynAmp[diagsPhys],
            IncomingMomenta -> momenta, List -> False] /. EFTOrder;
        ampPhys = EFTSeries[ampPhys /. expandPropPhys, 2] // ExpandScalarProduct;
        ampPhys = EFTSeries[ampPhys /. perturbativeOrder, 2];
```

At this point, we just need to replace the analytical expressions `ampFull` and `ampPhys` with rational randomly-generated kinematics. To this end, we use the Mathematica package `SpinorHelicity4D` [44]. Note that there are some incompabilities between `FeynArts`, `FeynCalc` and `SpinorHelicity4D`, so that loading everything from the very beginning can be troublesome. Indeed, when loading `SpinorHelicity4D` there may be some warnings for some variables present in more than one context, although this should not compromise the proper working of the notebook since we only need the functionalities declared from `SpinorHelicity4D`. In any case, it could be a good idea to save both amplitudes in an auxiliary file (better excluding the contexts of `FeynArts` and `FeynCalc`) and later load the expressions into another notebook. We then initialize `SpinorHelicity4D` and declare some necessary mathematical objects, such as the totally antisymmetric tensor, the Pauli matrices and the product in the Minkowski space.

```
In[15]:= << SpinorHelicity4D`
        eps = {{0, 1}, {-1, 0}};
        sigmas = {IdentityMatrix[2], {{0, 1}, {1, 0}}, {{0, -I}, {I, 0}},
            {{1, 0}, {0, -1}}};
        MDot[a_List, b_List] := a[[1]]b[[1]] - a[[2;;-1]].b[[2;;-1]];
```

To generate spinors, we use the `GenSpinor` function from the `SpinorHelicity4D` package. This function takes as input the four-momenta, assuming all have the same mass, and outputs a list of two pairs of spinors for each massive momentum: $\lambda_\alpha, \widetilde{\lambda}^{\dot\alpha}, \mu_\alpha, \widetilde{\mu}^{\dot\alpha}$. The function returns a list where each element corresponds to spinors associated to momenta with the same mass. To extract the spinors in our case, we assign the first element of this list to `spScalars`. Finally, we compute the numerical value of the mass squared (`m2`) for the generated momenta, which is then used to normalize all momenta to a symbolic mass, `mS`.

```
In[16]:=  scalars = Table["s" <> ToString[i], {i, 1, nScalars}];
          spinors = GenSpinors[scalars, SameMasses -> scalars,
              DisplaySpinors -> True, ParameterRange -> 50];
          spScalars = spinors[[1]];
          m2 = S["s1"] // ToNum;

In[17]:=  momScalars = ( KroneckerProduct[eps.#[[1]], #[[2]]] +
              m2/(#[[1]].eps.#[[3]] #[[2]].eps.#[[4]])
              KroneckerProduct[eps.#[[3]], #[[4]]] ) &/@ spScalars;
          psScalars = Table[mS/(2 Sqrt[m2]) Tr[#.sigmas[[j]]], {j,4}] &/@ momScalars;
```

After computing the numerical kinematics using the formulas provided in Appendix A, we define a replacement rule to incorporate the randomly generated kinematic variables into the amplitudes. At this stage, we also substitute the parameter `mphys2` and split the resulting expressions into contributions with different EFT orders. Then, this procedure yields three equations: the first one being trivial since it corresponds to EFT order 4 and other two corresponding to the dimension-6 and dimension-8 contributions, respectively.

```
In[18]:=  listReplacements =
              Table[Momentum[momenta[[i]]] -> psScalars[[i]], {i, nScalars}];

In[19]:=  ampRedBasis = ampFull //. listReplacements //. {Pair -> MDot} // Expand;
          ampRedBasis = CoefficientList[
              EFTSeries[ampRedBasis /. {Power[mS, n_] :> mphys2^(n/2)}, 2], invL2];

In[20]:=  {ampReddim6, ampReddim8} = {ampRedBasis[[2]], ampRedBasis[[3]]};

In[21]:=  ampPhysBasis = ampPhys //. listReplacements //. {Pair -> MDot} // Expand;
          ampPhysBasis = CoefficientList[
              EFTSeries[ampRedBasis /. {Power[mS, n_] :> mphys2^(n/2)}, 2], invL2];
          {ampPhysdim6, ampPhysdim8} = {ampPhysBasis[[2]], ampPhysBasis[[3]]}

In[22]:=  AppendTo[equationsdim6, ampReddim6 == ampPhysdim6];
          AppendTo[equationsdim8, ampReddim8 == ampPhysdim8];
```

The steps from `In[16]` to `In[22]` must be repeated using the Do loop function to generate enough equations to determine all the desired coefficients. Here, of course, variables `equationsdim6` and `equationsdim8` need to have been previously initialized to an empty list {} before the Do statement.

Finally, we solve both algebraic systems of linear equations to determine the coefficients that are to be shifted in the Lagrangian in the physical basis. We begin by solving the dimension-6 equations and then substitute their solution into the dimension-8 equations to obtain the remaining contributions. Combining `soldim6` and `soldim8` we obtain the results in Eqs. (18)-(22).

```
In[23]:=  coefphysdim6 = {lmbdphysdim6, a61physdim6};
          soldim6 = Flatten@Solve[equationsdim6, coefphysdim6] // Expand;
          coefphysdim8 = {lmbdphysdim6, a61physdim8, a81physdim8, a82physdim8};
          soldim8 = Flatten@Solve[equationsdim8 /. soldim6, coefphysdim8] // Expand;

In[24]:=  mphys2
          lmbdphys /. perturbativeOrder /. soldim6 /. soldim8
```

```
        a61phys /. perturbativeOrder /. soldim6 /. soldim8
        a81phys /. perturbativeOrder /. soldim6 /. soldim8
        a82phys /. perturbativeOrder /. soldim6 /. soldim8
```

Out[24]= m0$^2$ - 2 invL2 m0$^4$ r61 + 2 invL2$^2$ (4 m0$^6$ r61^2 + m0$^6$ r81)

Out[25]= lmbd + invL2 m0$^2$ ( r62 - 8 r61 lmbd)
        +invL$^2$ m0$^4$ (-10 r61 r62 - r82 - r83 + 64 r61$^2$ lmbd + 12 r81 lmbd)

Out[26]= a61 - 4 r62 lmbd + 16  r61 lmbd$^2$ + invL2 m0$^2$ (-12 a61 r61 - $\frac{22}{5}$ r62$^2$ - r84
        + $\frac{512}{5}$ r61 r62 lmbd + $\frac{56}{5}$ r82 lmbd + 8 r83 lmbd - $\frac{1728}{5}$ r61$^2$ lmbd$^2$
        - $\frac{304}{5}$ r81 lmbd$^2$)

Out[27]= a81 + 6 a61 r62 - 48 a61 r61 lmbd - $\frac{108}{5}$ r62$^2$ lmbd - 4 r84 lmbd
        + $\frac{1248}{5}$ r61 r62 lmbd$^2$ + $\frac{144}{5}$ r82 lmbd$^2$ + 16 r83 lmbd$^2$ - $\frac{3072}{5}$ r61$^2$ lmbd$^3$
        - $\frac{576}{5}$r81 lmbd$^3$

Out[28]= a82

   This code shows how we can perform on-shell matching in a rather easy manner. Indeed, despite the drawback of the number of diagrams to be considered in on-shell matching, we can actually side-step this issue by matching just a few number of amplitudes (or even one) with many external legs. This way, as discussed in Sec. 3.1, we have been able to perform the full reduction of the real scalar singlet up to dimension 8 using only the 8-point amplitude. The full code is provided as an ancillary file in the arXiv submission of this manuscript.

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
