# Peer review of "Efficient on-shell matching"

_SciPost Physics, doi:SciPost Phys. 18, 185 (2025)_

## Round 1 · Referee Report · Anonymous (Referee 1) · 2025-1-9

Report

In this article, the authors present an efficient algorithm for the on-shell matching of effective field theories (EFTs). While the off-shell matching has been automated in recent years, no efficient method for on-shell matching has been previously available. The central advantage of this approach, which is presented by the authors, is that it allows to perform the matching directly in a minimal basis of the EFT, and no operators that are redundant due to field redefinitions have to be considered, contrary to the off-shell approach. This offers the potential to significantly simplify various EFT calculations, as the authors show in several examples.

The paper is very pedagogical and all details are clearly presented. I only have a few minor comments as suggestions to the authors:

1) While the treatment of evanescent operators is excellently described in section 2.1, the introduction could give the impression that the issues of evanescent operators are circumvented by performing the matching on-shell. However, these issues are resolved by computing also the relevant parts of the soft region in the EFT and full theory. To my understanding, this could equally well be done in an off-shell matching calculation to remove the evanescent structures. I would thus suggest, to add some clarification in the introduction, stating that the removal of evanescent operators is due to computing the soft region and not due to matching on-shell.

2) In the conclusions, it is written "No redundant or evanescent operators have to be included in the calculation". I would suggest to exchange the word "calculation" with, e.g., "basis". In the intermediate steps of the calculation, i.e., when taking the difference of the soft regions of EFT and full theory, evanescent structures are present, even though their explicit form is never visible. While it is unavoidable to have these evanescent operators in the calculation, the advantage of the algorithm presented in this work is that their form never has to be specified and their physical effects are automatically computed.

3) In the context of the matching conditions in Eqs. (65-74), it might be useful to indicate for completness that the matching scale is chosen as the mass of the heavy Higgs $(\mu_\text{match}=M)$.

4) In appendix A, the authors mention the algorithm used for generating the rational on-shell kinematics. I belief it would be valuable to add the corresponding references [54,55] also in the main part of the paper, where appropriate, and not only in the appendix.

5) When listing the higher-dimensional EFT operator bases for SMEFT/LEFT in the introduction, the authors should consider also adding the following citation Li et al. [2005.00008] for the $d=8$ SMEFT basis.

6) In the second to last line on page 22 there is presumably a typo in the variable name $\texttt{perturbatuveOrder}$.

Recommendation

Ask for minor revision

---

## Round 2 · Referee Report · Anonymous (Referee 2) · 2025-1-14

Strengths

Accept

Weaknesses

None

Report

Accept for publication

Recommendation

Publish (surpasses expectations and criteria for this Journal; among top 10%)

---

## Round 2 · Referee Report · Anonymous (Referee 1) · 2025-1-17

Report

I thank the authors for addressing all points previously raised and for their modifications of the manuscript, which is of very high quality.
I recommend it for publication in SciPost Physics.

Recommendation

Publish (surpasses expectations and criteria for this Journal; among top 10%)

---

## Round 2 · Referee Report · Anonymous (Referee 3) · 2025-5-6

Report

The paper targets a very interesting an timely problem in all of particle physics - the balance between general EFT approaches and UV-completions. While in general the matching of theories is a solved problem, the stack of SM-related effective theories with the intermediate symmetry breakings make it very hard in practice. Moreover, in the precision-LHC era, the matching has to be performed beyond leading order, which requires advanced numerical tool.

For this paper, I am not sure what happened in the past. I read the one substancial referee report and the response of the authors, it seems all fine to me.

One additional aspect which I am missing, through, is at least a comment on specific use cases. There are papers where different groups compare analyses in EFT and full UV-frameworks, usually to D6 and D8. What would change for these published analyses with this new approach? Providing this kind of context would very much help the non-expert readers or potential users.

After adding a brief discussion along these lines, I think the paper should be published.

Recommendation

Publish (easily meets expectations and criteria for this Journal; among top 50%)

  • validity: -
  • significance: -
  • originality: -
  • clarity: -
  • formatting: -
  • grammar: -

Author:  Mikael Chala  on 2025-05-08  [id 5468]

(in reply to Report 3 on 2025-05-06)
Category:
remark

We would like to thank the referee for his/her comments on our manuscript (incidentally, we don't understand either what the problem with the previous reports was). Regarding the referee's request, our method is completely equivalent to the one previously used based on off-shell matching in terms of the results obtained with them. They are just two different ways of computing the same thing. The difference is that our method can be much more convenient than the traditional one in certain circumstances. For instance, the full reduction of the SMEFT Green's basis to the physical one at mass dimension 8 has not been completed yet, despite the fact that the two bases have been known for five years. The reason is that the reduction is extremely challenging and some of the available partial results could only be completed using our new on-shell matching method.

---

## Round 2 · List of Changes

Dear Editor,

We would like to thank the referee for the positive feedback as well as for the concrete suggestions, which we have addressed as follows.

  1. The treatment of evanescent operators is indeed identical in both on-shell and off-shell matching, and the standard procedure for computing the evanescent shifts in the on-shell case has been discussed in our current Ref. [41] (Ref. [40] in the previous submission). The difference in our particular case is that, due to our numerical procedure, we have to do the matching in $d=4$ and therefore have to generate the evanescent structures via the soft region expansion in the EFT. The same could be also done in the off-shell matching.

We have added a new footnote in page 3 to account for this.

  1. We have followed the referee's recommendation; see page 18.

  2. We have added a comment before equation 65.

  3. We have added a comment including these references in page 4.

  4. We are sorry for having forgotten this reference, which we have now included.

  5. We thank the referee for spotting this typo, which is now fixed.

Moreover, we have fixed a typo in Figure 4 and equation 45 ($H^4D^2$ should read $H^4 D^4$).

All modifications in the text are shown in red colour.

Thank you very much.

Best wishes, The authors.

---

## Editorial Decision

published